# ApoA-I Nanoparticles as Curcumin Carriers for Cerebral Endothelial Cells: Improved Cytoprotective Effects against Methylglyoxal

**DOI:** 10.3390/ph15030347

**Published:** 2022-03-13

**Authors:** Sai Sandhya Narra, Sarah Rosanaly, Philippe Rondeau, Jessica Patche, Bryan Veeren, Marie-Paule Gonthier, Wildriss Viranaicken, Nicolas Diotel, Palaniyandi Ravanan, Christian Lefebvre d’ Hellencourt, Olivier Meilhac

**Affiliations:** 1INSERM UMR 1188, Diabète athérothombose Réunion Océan Indien (DéTROI), Université de La Réunion, 97490 Saint-Denis, La Réunion, France; narra.sai-sandhya@univ-reunion.fr (S.S.N.); sarah.rosanaly@univ-reunion.fr (S.R.); philippe.rondeau@univ-reunion.fr (P.R.); jessica.patche@univ-reunion.fr (J.P.); bryan.veeren@univ-reunion.fr (B.V.); marie-paule.gonthier@univ-reunion.fr (M.-P.G.); nicolas.diotel@univ-reunion.fr (N.D.); christian.lefebvre-d-hellencourt@univ-reunion.fr (C.L.d.H.); 2Unité Mixte Processus Infectieux en Milieu Insulaire Tropical (PIMIT), Université de La Réunion, INSERM UMR 1187, CNRS UMR 9192, IRD UMR 249, Plateforme Technologique CYROI, 94791 Sainte Clotilde, La Réunion, France; wildriss.viranaicken@univ-reunion.fr; 3Department of Microbiology, School of Life Sciences, Central University of Tamil Nadu, Thiruvarur 610005, India; ravanan@cutn.ac.in; 4Centre Hospitalier Universitaire (CHU) de La Réunion, CIC 1410, 97410 Saint-Pierre, La Réunion, France

**Keywords:** nanoparticle, curcumin, HDL, methylglyoxal, cerebral endothelial cells, endothelial dysfunction

## Abstract

Methylglyoxal (MGO) is a highly reactive metabolite of glucose present at elevated levels in diabetic patients. Its cytotoxicity is associated with endothelial dysfunction, which plays a role in cardiovascular and cerebrovascular complications. Although curcumin has many therapeutic benefits, these are limited due to its low bioavailability. We aimed to improve the bioavailability of curcumin and evaluate a potential synergistic effect of curcumin and reconstituted high-density lipoprotein (rHDL) nanoparticles (Cur-rHDLs) on MGO-induced cytotoxicity and oxidative stress in murine cerebrovascular endothelial cells (bEnd.3). Cur-rHDL nanoparticles (14.02 ± 0.95 nm) prepared by ultracentrifugation and containing curcumin were quantified by LC–MS/MS. The synergistic effect of cur-rHDL nanoparticles was tested on bEnd.3 cytotoxicity, reactive oxygen species (ROS) production, chromatin condensation, endoplasmic reticulum (ER) stress, and endothelial barrier integrity by impedancemetry. The uptake of curcumin, alone or associated with HDLs, was also assessed by mass spectrometry. Pretreatment with Cur-rHDLs followed by incubation with MGO showed a protective effect on MGO-induced cytotoxicity and chromatin condensation, as well as a strong protective effect on ROS production, endothelial cell barrier integrity, and ER stress. These results suggest that Cur-rHDLs could be used as a potential therapeutic agent to limit MGO-induced dysfunction in cerebrovascular endothelial cells by enhancing the bioavailability and protective effects of curcumin.

## 1. Introduction 

Type 2 diabetes (T2D) is a global health issue that increases the risk of cardiovascular and cerebrovascular diseases, including ischemic stroke and its hemorrhagic complications. Prolonged hyperglycemia leads to various biochemical modifications, such as oxidative stress, apoptosis, and glycation, particularly mediated by methylglyoxal (MGO), a physiological reactive carbonyl compound derived from glucose metabolism. High concentrations of MGO and MGO-derived products are found in the plasma of diabetic patients and are associated with diabetic complications [1,2,3]. Increased plasma MGO levels have been shown to be associated with cardiovascular disease and mortality in T2D patients [4]. MGO induces massive intracellular oxidative stress, particularly in endothelial cells (ECs) at the interface between blood and different tissues. This reactive compound is a precursor of advanced glycation products (AGEs) but has a greater potential than AGEs or glucose itself to induce vascular damage [5]. MGO has been shown to be cytotoxic to different types of ECs, including human umbilical vein ECs (HUVECs) and cerebral ECs [6,7]. It was shown to induce endoplasmic reticulum stress (ER stress) that may lead to apoptosis [5]. To prevent MGO- or high-glucose-induced oxidative stress and resulting cytotoxicity, various antioxidants have been used, such as proanthocyanidins [8], isorhamnetin [7], resveratrol [9], and pterostilbene, a natural derivative of resveratrol [10]. Curcumin, a potent but hydrophobic antioxidant, anti-inflammatory, and anti-apoptotic compound, has been used to limit MGO-induced cytotoxicity in different cell types such as mouse embryonic stem cells [11] and human hepatoma cells [12] but not in ECs. In the present study, we sought to test the protective effect of curcumin on MGO-induced intracellular oxidative stress and cytotoxicity in cerebral ECs. We also tested the capacity of curcumin to limit the increase in endothelial permeability induced by MGO by monitoring real-time cell impedance. Despite its numerous neuroprotective effects, curcumin has limited bioavailability and poor solubility. Our second objective was to improve curcumin bioavailability by loading it into Apolipoprotein A-I phospholipid nanoparticles (also called reconstituted high-density lipoproteins-rHDLs) and test the potential synergistic effect of curcumin and HDLs versus HDLs or curcumin alone. HDL particles themselves have pleiotropic properties similar to those of curcumin (antioxidant, anti-inflammatory, and anti-apoptotic effects). In particular, HDL treatment limits infarct volume and mortality in different models of ischemic stroke via endothelial protective effects [13,14]. In the present study, we tested a potential synergistic effect of rHDLs and curcumin on MGO-induced oxidative stress, endothelial permeability, and cytotoxicity in cerebral endothelial cells. 

## 2. Results

### 2.1. Particle Size Distribution of rHDL and Cur-rHDLs by DLS Analysis

Dynamic light scattering was used to determine the particle size distribution of rHDLs and Cur-rHDLs. The particle average size was determined to be 12.55 ± 0.21 nm for rHDLs and 14.02 ± 0.95 nm for Cur-rHDLs, as shown in Figure 1a,b respectively. A polydispersity index (PDI) near to zero implies homogeneity of dispersions, and a PDI greater than 0.3 indicates heterogeneity [15]. The PDI of our Cur-rHDL nanoparticles was around 0.25. The incorporation of curcumin did not significantly affect the size of the rHDL particles (*p* > 0.05, comparing Cur-rHDL to rHDL particle size).

### 2.2. Effect of HDL and Curcumin on Cerebrovascular Endothelial Cell Cytotoxicity Induced by MGO

MGO is known to be cytotoxic to endothelial cells, but MGO cytotoxicity has never been evaluated in bEnd.3 cells [4,7]. Cell viability was determined by the MTT assay after incubation with different concentrations of MGO for 24 h. MGO showed concentration-dependent cytotoxicity with decreased cell viability of approximately 20% and 70% at 1 mM and 2 mM, respectively (Figure 2a). These results are in agreement with previously published results of MGO cytotoxicity [16]. Next, we investigated the effect of rHDLs (H) and curcumin (C) on MGO cytotoxicity in cerebral endothelial cells by pretreating the cells with different concentrations of rHDLs and curcumin for 1 h and then adding 2 mM MGO for 24 h. Reconstituted HDLs were not cytotoxic but showed a cytoprotective effect on MGO-induced cell death in a concentration-dependent manner. While low rHDL concentrations of 5–50 μg/mL were not cytoprotective, higher concentrations ranging from 100 to 200 μg/mL were significantly cytoprotective (Figure 2b). On the other hand, curcumin started to be protective at 0.2 and 0.4 μM but was not effective at 0.1 μM and was potentially cytotoxic at higher doses (0.8, 1m and 2 μM), with a loss of the cytoprotective effect (Figure 2c).

### 2.3. Effect of Curcumin-Enriched rHDLs on MGO Cytotoxicity in Cerebral Endothelial Cells

Given that rHDLs (H) and curcumin (C) alone showed a cytoprotective effect, we tested the synergistic effect of curcumin and rHDLs (Cur-rHDL) on the cytotoxicity of MGO. For this purpose, the cells were pre-treated with rHDLs (H), curcumin (C), or curcumin-enriched rHDLs (Cur-rHDL) for 1 h before the addition of 2 mM MGO. Cur-rHDLs contained 50 to 80 μg/mL ApoA-I and 0.03 to 0.048 μM curcumin, concentrations that were not protective when rHDLs and curcumin were used alone to prevent MGO cytotoxicity. At these concentrations, Cur-rHDLs were not cytotoxic. The cytoprotective effect was not observed at a concentration of 50 μg/mL rHDL enriched with 0.03 μM curcumin either alone or in combination (Figure 3a). In contrast, concentrations of 80 μg/mL rHDL and 0.048 μM curcumin showed a synergistic cytoprotective effect compared to rHDL and curcumin alone in the presence of MGO (Figure 3b). Similarly, 100 μg/mL of rHDLs enriched with 0.06 μM curcumin and 200 μg/mL of rHDLs enriched with 0.12 μM curcumin showed an enhanced cytoprotective effect relative to curcumin and rHDLs alone, which were already cytoprotective at these concentrations (Figure 3c,d respectively). 

### 2.4. Effect of Curcumin-Enriched rHDLs on Cerebral Endothelial Layer Integrity

To investigate the integrity of the endothelial layer, the cells were treated with different concentrations of MGO (0.8, 1, and 2 mM), and the cell real-time electrical impedance was measured using the xCELLIgene system. Cerebral endothelial cell integrity was not affected by MGO at 0.8 mM and 1 mM but was altered at 2 mM, resulting in a decreased impedance compared to the control (Figure 4a). This is consistent with the cytotoxic effect of MGO at 2 mM observed by the MTT assay (Figure 2a). We further tested the effect of curcumin-enriched rHDLs on MGO-induced reduction of impedance. These results showed a synergistic protective effect on the integrity of the cerebral endothelial cell monolayer impaired by MGO, whereas individually, rHDLs and curcumin were not protective (Figure 4b). This suggests that curcumin-enriched rHDLs contribute to protecting the integrity of cerebral endothelial cells from the cytotoxic effects of MGO.

### 2.5. Effect of Curcumin-Enriched rHDLs on Intracellular ROS Production Induced by MGO in Cerebral Endothelial Cells 

We then evaluated the effect of curcumin-enriched rHDLs on intracellular reactive oxygen species (ROS) production in the presence of MGO by DCFH-DA fluorimetry. The cells were pretreated with rHDLs, curcumin, and Cur-rHDLs for 1 h before the addition of 2 mM MGO from 1 to 6 h. MGO increased the intracellular ROS production as early as 2 h, which peaked at 3 h and started to decrease slightly at 4–6 h, while remaining above that in the control. Curcumin-enriched rHDLs limited significantly MGO-induced ROS production compared to rHDLs and curcumin alone (except for rHDL at 5 h that showed a significant ROS reduction) (Figure 5).

### 2.6. Effect of Curcumin-Enriched rHDLs on MGO-Induced Chromatin Condensation in Cerebral Endothelial Cells

We then investigated the effect of curcumin-enriched rHDLs on MGO-induced chromatin condensation in cerebral endothelial cells by DAPI staining. MGO induced chromatin condensation that was significantly reduced by pre-treatment with curcumin-enriched rHDLs, whereas rHDLs and curcumin alone showed no significant protective effect (Figure 6a,b).

### 2.7. Effect of Curcumin-Enriched rHDLs on MGO-Induced ER Stress in Cerebral Endothelial Cells

The effect of curcumin-enriched rHDLs on MGO-induced ER stress was assessed by immunofluorescence. During ER stress, Xbp-1 and ATF-4 are translocated into the nucleus, whereas GRP-78 levels increase. Nuclear translocation of Xbp-1 and ATF-4, as well as increased levels of GRP-78, were observed upon MGO exposure, similar to what seen in the presence of thapsigargin, which is a well know ER stress inducer. Pre-treatment with rHDLs or curcumin alone did not have a significant effect on these ER stress markers, whereas curcumin-enriched rHDLs prevented the nuclear translocation of Xbp-1 (Figure 7a) and ATF-4 (Figure 7b) and decreased the levels of GRP-78 (Figure 7c). This suggests that curcumin-enriched rHDLs have a protective effect on MGO-induced ER stress.

### 2.8. Quantification of Cellular Curcumin after Incubation of bEend 3 Cells with Curcumin-Enriched rHDLs or Curcumin Alone

We evaluated the cellular uptake of free curcumin compared to that of Cur-rHDLs by bEnd 3 cells to understand the impact of encapsulation on curcumin uptake. The uptake of curcumin was assessed by mass spectrometry. bEnd.3 cells were incubated with free curcumin or curcumin-enriched rHDLs for 3 and 6 h. The intracellular uptake of free curcumin was faster and higher than that of curcumin-enriched rHDLs at both 3 and 6 h. The uptake of free curcumin was similar at 3 and 6 h, whereas curcumin-enriched rHDL uptake was greater at 3 h and decreased at 6 h, suggesting saturation of the HDL uptake process (Figure 8). 

## 3. Discussion

Most diseases involve an imbalance between free radical and antioxidant responses leading to oxidative stress. This is notably the case in diabetes and its cardiovascular and cerebrovascular complications. Methylglyoxal (MGO) is a highly reactive metabolite of glucose metabolism whose levels are increased in type II diabetic patients. It has been shown to accumulate in the brain of mice deficient for glyoxalase 1/vitamin B6 (MGO detoxification system), suggesting that this imbalance is particularly critical for MGO accumulation in the brain [17]. Studies have shown that the dysfunction of the glyoxalase system is related to several diabetic complications, including macrovascular diseases in humans [18]. As vascular dysfunction plays an important role in diabetes-associated vascular complications, several studies have been conducted to understand the effect of MGO on endothelial dysfunction [19,20]. Many antioxidants are used to reduce MGO cytotoxicity, such as resveratrol, curcumin, and luteolin, but their use is limited by their poor solubility and bioavailability. Improving the bioavailability of antioxidant molecules to reach the cells or tissues of interest and their therapeutic benefits in the body remains a topic of intense research. 

The present study aimed to test the ability of curcumin and rHDLs to limit MGO-induced oxidative stress, cytotoxicity, and endoplasmic reticulum (ER) stress in murine cerebral endothelial cells (bEnd 3). In a second step, the potential synergistic effect of curcumin- and rHDLs, which could improve curcumin bioavailability, was also evaluated on the same processes. For this purpose, rHDLs were enriched with curcumin and then re-isolated by ultracentrifugation to remove unincorporated free curcumin. The average particle size of Cur-rHDLs was determined to be 14.02 ± 0.95 nm. MGO was cytotoxic (2 mM) in our cerebral endothelial cell model and, previously, was also shown to be cytotoxic in different cell lines including neuroglial cells, HUVECs, and human brain endothelial cells [7,21,22]. MGO also induced reactive oxygen species (ROS) formation [23,24], blood–brain barrier (BBB) hyperpermeability [25,26], chromatin condensation, and ER stress [27,28], as shown in previous studies. 

MGO damage and cytotoxicity in endothelial cells (ECs) is mediated by oxidative stress and reactive oxygen species (ROS) formation [24]. Previous studies have shown that many compounds reduce MGO cytotoxicity in endothelial cells via reducing ROS formation. MGO is known to increase ROS production, associated with the weakening of barrier integrity. MGO was shown to promote oxidative stress in brain endothelial cells [16]. 

Previous studies have shown that MGO leads to increased permeability of brain endothelial cells. This study reported that the decrease of transendothelial electrical resistance by MGO was associated with alterations in tight junction proteins, such as glycation of occludin and redistribution of zonula occludens-1 (ZO-1). MGO treatment also induced the redistribution of claudin-5 and of the adherens junction protein β-catenin in brain endothelial cells [25]. Claudin-5 is the major regulatory protein of BBB permeability [29]. MGO induced the activation of UPR signaling and apoptosis in HUVECs via AMPK, PI3K–AKT, and CHOP induction pathways [30,31]. The mechanism of ER stress inhibition by curcumin is not well known but it has been shown to reduce ER stress responses by increasing peroxiredoxin 6 (prdx 6), decreasing NF-ĸB signaling, and reducing oxidative stress [32,33].

The curcumin metabolites, glucuronides and di-tetra- and hexahydrocurcumins were detected in the liver, kidney, and intestines after oral, intraperitoneal (IP), and intravenous (IV) administrations. The oral administration of curcumin showed reduced bioavailability and poor absorption compared to IV or IP administrations [34]. The IP administration of curcumin allowed a wide biodistribution including to the brain, lungs, liver, and kidney. Nanoemulsions of curcumin are reported to improve the bioavailability and therapeutic effects of this molecule compared to free curcumin in both in vitro and in vivo studies [35]. 

Interestingly, our curcumin-enriched rHDL (Cur-rHDLs) nanoparticles showed an improved synergistic protective effect on MGO-induced cytotoxicity, ROS production, cerebral endothelial cell integrity, chromatin condensation, and ER stress compared to rHDLs or curcumin alone. Anti-apoptotic effects of curcumin in MGO-stimulated endothelial cells have been previously reported. The authors suggested potential direct scavenging of MGO by curcumin and one of its analogues, dimethoxy curcumin [36]. However, the protection by HDLs from MGO-induced endothelial cell death had never been reported before. Here, we provide the first evidence that reconstituted HDL particles exhibit sufficient antioxidant and anti-inflammatory effects to prevent MGO-induced cell death in endothelial cells. 

Recently, the bioavailability of curcumin was improved using nanoemulsions administered intraperitoneally in a mouse model of Parkinson’s disease [36]. Poloxamer nanoparticles (nonionic triblock copolymers) displayed better penetration of the blood–brain barrier and better accumulation in the brain than non-encapsulated curcumin in subjects with Alzheimer’s disease [37]. Previous studies on curcumin encapsulated in low-density lipoprotein (LDL) and HDL particles showed that LDLs were more prone to accumulate curcumin than HDL particles and highlighted their potential to treat cancer, due to the avidity of cancer cells for LDLs [38].

A previous study showed that HDLs reconstituted with soy phosphatidylcholine and Apolipoprotein A-I isolated from the plasma of healthy volunteers significantly reduced the infarct area in two rat models of stroke and had an antioxidant effect in human endothelial and neuroblastoma cell lines [39]. The HDL drug delivery system facilitates the cellular uptake of drugs via interaction with scavenging receptors (SR-BI) by bypassing the endosomal/lysosomal pathway, independently of HDL uptake by transcytosis. Cerebral endothelial cells (including bEnd.3) contain SR-BI class B type I receptors in their membranes. rHDLs injected intraperitoneally were detected in the liver and kidney in in vivo mouse and zebrafish models [40]. rHDLs were also shown to be taken up by endothelial cells and astrocytes in the brain of mice subjected to experimental ischemic stroke. We recently demonstrated that HDLs isolated from plasma had a neuroprotective effect via SR-BI (scavenger receptor type BI) in a mouse model of MCAO [13]. Improvement of HDL potential to prevent the deleterious effects of stroke in diabetic conditions is of major importance, since our preliminary results tend to demonstrate that HDLs alone are not sufficient to limit infarct volume and hemorrhagic complications in a mouse model of stroke under hyperglycemic conditions [41]. Curcumin-enriched HDLs could then represent a therapeutic option for brain pathologies involving oxidative stress and BBB dysfunction. Nano-curcumin administered orally was shown to have a good brain biodistribution in a mouse model of experimental cerebral malaria [13]. 

## 4. Materials and Methods

Methylglyoxal (MGO), curcumin (Cat. Number: C7727), propidium iodide (PI), thiazolyl blue tetrazolium bromide (MTT), and other chemicals were purchased from Sigma Aldrich (St. Louis, MO, USA). 

### 4.1. Cell Culture

The murine cerebral endothelial cell line bEnd.3 was obtained from the American Type Culture Collections (ATCC^®^ CRL^™^-2299™, Manassas, VA, USA). The cells were cultured in Dulbecco’s Modified Eagle Medium (DMEM) containing 25 mM glucose, 10% heat-inactivated Fetal Bovine Serum (FBS), 5 mM L-glutamine, 2 µg/mL streptomycin, and 50 µU/mL penicillin (Pan Biotech, Dutscher, and Brumath, France). The cells were kept in a humidified atmosphere with 5% CO2, at 37 °C. All the treatments were carried out in serum-free medium.

### 4.2. Preparation of Curcumin-Enriched rHDLs (Cur-HDLs)

Reconstituted HDLs (rHDLs) were obtained from CSL Behring AG (CSL111, Bern, Switzerland). rHDLs enriched with curcumin were prepared by gently mixing 3 mL of 2 mg/mL rHDLs with 5 μM curcumin followed by incubation in the dark for 16 h at 37 °C (50 rpm). Curcumin-enriched rHDLs (Cur-rHDLs) were isolated by ultracentrifugation as described previously [13] to eliminate free curcumin. Briefly, the rHDL curcumin mixture was adjusted to a density d = 1.22 with KBr and overlaid with a KBr saline solution (d = 1.21). Ultracentrifugation was performed at 100,000× *g* for 24 h at 10 °C. The curcumin-enriched rHDL yellow fraction (top layer) was recovered. The curcumin-enriched rHDL fraction was desalted by centrifugation with PBS 5 times, 20 min each, at 12,000 × *g* using a Centricon filter (cutoff 10 kDa; Vivascience, Stonehouse, UK) to remove the excess of KBr and concentrate the sample. Protein concentration was determined using the Bicinchoninic Acid (BCA) method (BCA Protein Assay Kit, Thermo Scientific Pierce, and Waltham, MA, USA).

### 4.3. Quantification of Curcumin by LC–MS Analysis

#### 4.3.1. Extraction of Curcumin from HDL-Curcumin

Curcumin was extracted from Cur-HDLs by precipitation of ApoA-I with acetonitrile (ACN). In brief, 54 µL of ACN was added to 27 µL of the Cur-HDL solution (2:1, *v*/*v*), vortexed a few seconds, and centrifuged at 20,000 × *g* for 15 min. The resulting supernatant was collected and directly analyzed by mass spectrometry.

#### 4.3.2. Identification and Quantification of HDL-Curcumin by LC–MS/MS

Curcumin from Cur-HDLs was identified by ultra-high-performance liquid chromatography coupled with a HESI-Orbitrap mass spectrometer (Q Exactive™ Plus, Thermo Fisher). Briefly, 10 µL of sample was injected in a UHPLC system equipped with a Thermo Fisher Ultimate 3000 series WPS-3000 RS autosampler and then separated on a PFP column (2.6 μm, 100 mm × 2.1 mm, Phenomenex, Torrance, CA, USA). Elution was performed using a binary gradient of water and ACN, both acidified with 0.1% formic acid (B). A flow rate of 450 µL/min was used. The following elution was setup: 0.0–3.0 min, from 20 to 50% B; continue with 50% B for 8 min. Then, the column was washed with 95% B for 3 min and equilibrated with 20% B for 2 min. The column oven was thermostated at 30 °C. 

For mass spectrometry conditions, the Heated Electrospray Ionization source II was set at 3.9 kV, the capillary temperature at 320 °C, the sheath gas flow rate at 65 units, the auxiliary gas flow rate at 20 units, and the S-lens RF at 50%. Mass spectra were registered in full scan data-dependent acquisition from *m*/*z* 100 to 500 in positive ion mode at a resolving power of 70,000 FWHM (at *m*/*z* 400). The automatic gain control (AGC) was set at 1e6, and the injection time (IT) at 200 ms. The MS/MS spectra were acquired at a resolving power of 17,500 FWHM (at *m*/*z* 200) with an AGC set to 2e5 and an IT of 50 ms. A relative higher energy collisional dissociation (HCD) energy of 40% was applied. Identification and quantification of curcumin were based on its retention time, accurate mass, elemental composition, MS fragmentation pattern, and comparisons with a commercial standard. Data were acquired by XCalibur 4.2.47 software (Thermo Fisher Scientific Inc. Waltham, MA, USA) and processed by Skyline 21.1.0.146 software (MacCoss Lab.). The performance of the Orbitrap was evaluated weekly, and external calibration of the mass spectrometer was performed before analysis with an LTQ ESI positive ion calibration solution (Pierce™). 

#### 4.3.3. Preparation of Standard Solution and Calibration Curve

The standard stock solution of curcumin was dissolved in DMSO at a concentration of 100 µM. Calibration standard solutions were prepared by dilution of the stock solution in 20% ACN acidified with 0.1% formic acid to obtain the desired calibration curves ranging from 0.0156 to 2 µM. The calibration curves were constructed by plotting the peak area of the analytes against the corresponding analyte concentrations with linear regression (1/x weighting, linear through zero) using standard samples at six concentrations. The obtained calibration curve had a correlation coefficient (R^2^) of 0.989. The concentration of curcumin from the Cur-HDLs was 2.8 ± 0.4 µM (Figure 9). 

### 4.4. Determination of Particle Siz

The particle size of rHDLs and Cur-rHDLs was determined using a dynamic light scattering (DLS) particle size analyzer (Zetasizer Nano, Malvern Instruments, UK), which measures the hydrodynamic diameter and polydispersity index (PDI) of the nanoparticles. The analysis was performed at 25 °C with a scattering angle of 173°. The rHDL and Cur-rHDL samples were prepared at a concentration of 1 mg/mL in PBS and filtered through a 0.22 µm-pore size PTFE syringe filter (to remove any aggregates). 

### 4.5. Evaluation of Cell Viability

Cell viability was evaluated by assessment of mitochondrial metabolic activity via the MTT assay (3-(4, 5-dimethylthiazol-2-yl)-2, 5-diphenyltetrazolium bromide) as previously described [42]. The cells were seeded in 96-well plates at a density of 5 × 10^3^ cells per well and grown for 24 h. The culture medium was removed, and the cells were washed once with PBS. The cells were then incubated with HDLs (H), curcumin (C), curcumin-enriched rHDLs (Cur-rHDLs) for 1 h before the addition of 2 mM MGO for 24 h. After 24 h, 20 μL of 5 mg/mL MTT reagent prepared in PBS was added to each well, and the cells were incubated in the dark at 37 °C for 2 h. Then, the medium was removed, the formed formazan crystals were dissolved in 100 μL DMSO, and the absorbance was measured at 570 nm (TECAN, Männedorf, Switzerland).

### 4.6. Evaluation of Intracellular ROS

The levels of intracellular reactive oxygen species (ROS) were measured using the fluorescent probe 2′,7′-dichlorodihydrofluorescein-diacetate (DCFH-DA) as described previously [42]. The cells were seeded in a black 96-well plate with a transparent bottom at a cell density of 5 × 10^3^ cells for 24 h. The medium was removed, and 100 μL of a 10 μM DCFH-DA solution in PBS was added to each well after washing the cells with PBS. After 40 min of incubation in the dark at 37 °C, the probe was removed and the cells were pre-treated with rHDLs (H), curcumin (C), curcumin-enriched rHDLs (Cur-rHDLs) for 1 h, before addition of 2 mM MGO for 1–6 h. ROS levels were measured by fluorescence analysis at an excitation wavelength of 492 nm and an emission wavelength of 520 nm (FLUOstar Optima, Bmg Labtech, and Cambridge, UK). 

### 4.7. Assessment of Endothelial Layer Integrity by Real-Time Electrical Impedance

The cellular integrity was analyzed by impedance measurement using the xCELLigence system (Acea Biosciences, San Diego, CA, USA). The cells were seeded in a 16-well xCELLigence plate specifically designed for monitoring the impedance, at a cell density of 4 × 10^3^ cells per well. After the cells reached confluency, they were pre-treated with rHDLs (H), curcumin (C), and curcumin-enriched rHDLs (Cur-rHDLs) for 1 h before the addition of 2 mM MGO. The impedance was monitored for 4 days.

### 4.8. Evaluation of Chromatin Condensation

Chromatin condensation was evaluated using the fluorescent dye DAPI (4′-6-diamidino-2-phenylindole). The cells were seeded on sterile 14 mm coverslips placed in a 24-well plate at a density of 5 × 10^3^ per well for 24 h. The culture medium was removed 24 h after the different treatment, and the cells were washed with PBS and fixed in 4% PFA for 10 min. The cells were then stained with DAPI (1 μg/mL) for 5 min and washed twice with PBS, and the coverslips were mounted on slides. The number of cells with condensed chromatin was determined using a fluorescent microscope (Eclipse 80i Nikon microscope equipped with a Hamamatsu digital camera, Life Sciences, Tokyo, Japan). The results are expressed as percentage of cells with condensed chromatin.

### 4.9. Immunocytochemistry 

Cells were seeded in a 24-well plate containing 14 mm-diameter coverslips at a density of 5 × 10^3^ per well for 24 h. After treatment, the cells were washed and fixed in 4% PFA (for ATF-4 primary antibody staining) for 5 min at room temperature or 100% ice-cold (−20 °C) methanol (for XBP-1 and GRP 78 primary antibodies staining) for 10 min and then incubated with the primary antibodies ATF-4 (Cell signaling technology-11815S, Danvers, MA; dilution 1:200–500 ng/mL), XBP-1 (Abcam-ab37152, Cambridge, UK; dilution 1:200–500 ng/mL), and GRP 78 (Abcam-ab21685, Cambridge, UK; dilution 1:500–200 ng/mL) prepared in blocking buffer (PBS containing 1% BSA) overnight at 4 °C. The primary antibody was removed, and the cells were washed 3 times with PBS. Secondary antibody and DAPI prepared in blocking buffer were added to the cells, which were incubated in the dark for 1 h at room temperature. The cells were washed with PBS, and the coverslips were mounted on slides that were examined under a confocal microscope (Nikon Eclipse Ti2 with a C2si confocal system, Tokyo, Japan). 

### 4.10. Intracellular Uptake of Curcumin-Enriched rHDLs

The cells were seeded in a 6-well plate at a density of 2 × 10^5^ for 24 h. The culture medium was removed, and cells were washed with PBS, incubated with rHDL (H), curcumin (C), and curcumin-enriched rHDLs (Cur-rHDLs) for 3 and 6 h. The medium was removed, and the cells were washed with 500 µL of PBS and then with 500 µL of 0.4% PBS–BSA/well to remove traces of polyphenols from the cell membrane. Then, 500 µL of methanol containing 200 mM HCl/well was added to collect the intracellular polyphenol moiety by cell scraping and transfer of the medium into Eppendorf tubes. After 45 min at 4 °C, the tubes were centrifuged at 14,000× *g* at 4 °C for 5 min, and the supernatant was collected for subsequent analysis by LC–MS/MS. For quantitation, chromatographic conditions and mass spectrometer parameters were the same as described above. A calibration curve ranging from 1.953 nM to 1000 nM was used. A quality control sample was analyzed within each batch, with two blank samples containing 20% ACN in water and 0.1% formic acid. 

### 4.11. Statistical Analysis

Data are expressed as mean ± SD of three independent experiments. Statistical analyses were performed by one-way analysis of variance (ANOVA) followed by Bonferroni’s multiple comparison test using Graph-Pad Prism 6 (Graph Pad Software, Inc., San Diego, CA, USA). Statistical significance was considered for a *p*-value ˂ 0.05. 

## 5. Conclusions

We demonstrated for the first time an improved protective effect of curcumin-enriched rHDL nanoparticles on MGO-induced cerebral endothelial dysfunction linked to oxidative stress and endoplasmic reticulum stress mechanisms, that need to be further investigated in vivo. These cur-rHDL nanoparticles are stable and biocompatible and have therapeutic potential as an emerging drug delivery system for pathologies involving endothelial dysfunction and oxidative stress, such as stroke under diabetic conditions.

## Figures and Tables

**Figure 1 pharmaceuticals-15-00347-f001:**
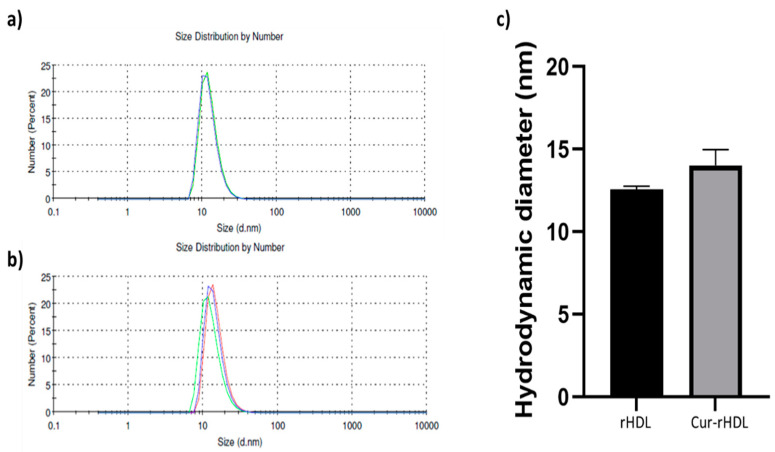
Particle size distribution of rHDLs and Cur-rHDLs determined by DLS analysis. The samples were prepared in PBS at 1 mg/mL. (**a**) Size distribution of rHDLs and (**b**) Cur-rHDLs and (**c**) Histograms representing the hygrodynamic particle diameter of rHDLs and Cur-rHDLs. Data are presented as the mean ± SD of three independent measurements for each sample.

**Figure 2 pharmaceuticals-15-00347-f002:**
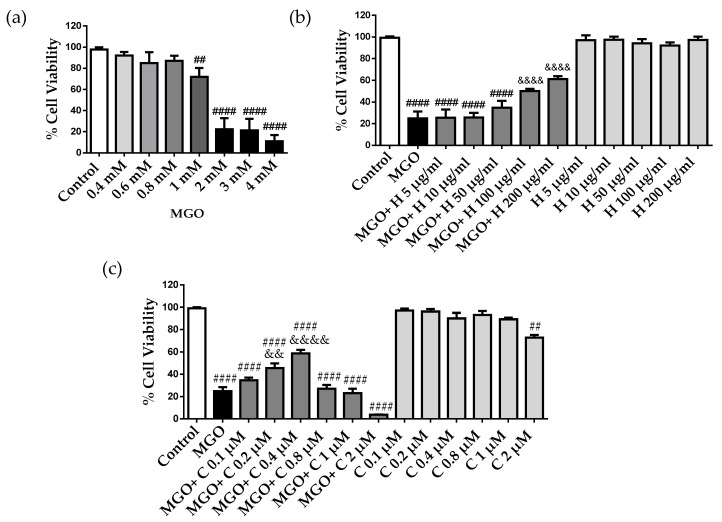
MGO cytotoxicity in bEnd3 cerebral endothelial cells, effect of rHDLs (H) and curcumin (C). (**a**) The cells were treated with different MGO concentrations (0.4–4 mM) for 24 h. (**b**) The cells were pre-treated with different concentrations of rHDLs, 5–200 μg/mL, (H) and (**c**) curcumin (C), 0.1–2 μM, for 1 h, followed by the addition of 2 mM MGO for 24 h. Cell viability was assessed by the MTT assay. Data are presented as mean ± SD of three independent experiments (*n* = 3). ## *p* ˂ 0.01, #### *p* ˂ 0.0001 as compared to control. && *p* ˂ 0.01, &&&& *p* ˂ 0.0001 as compared to MGO.

**Figure 3 pharmaceuticals-15-00347-f003:**
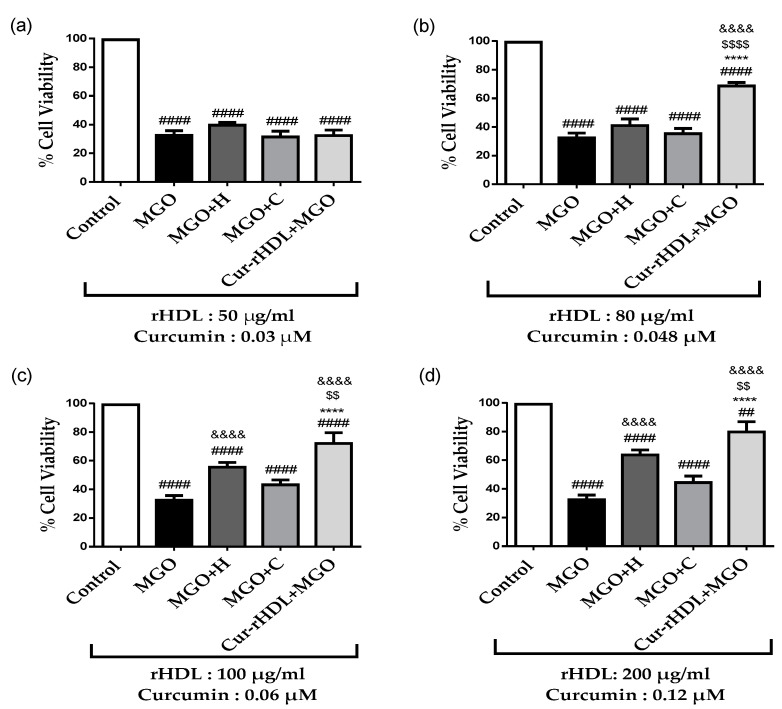
Effect of curcumin-enriched rHDLs on MGO cytotoxicity in cerebral endothelial cells. The cells were pre-treated with different concentrations of rHDLs (H), curcumin (C), and curcumin-enriched rHDLs (Cur-rHDLs) for 1 h before the addition of 2 mM MGO for 24 h. (**a**) H: 50μg/mL, C: 0.03 μM and Cur-rHDL: 50 μg/mL + 0.03 μM. (**b**) H: 80 μg/mL, C: 0.048 μM and Cur-rHDLs: 80 μg/mL + 0.048 μM. (**c**) H: 100 μg/mL, C: 0.06 μM and Cur-rHDLs: 100 μg/mL + 0.06 μM (**d**) H: 200 μg/mL, C: 0.12 μM and Cur-rHDLs: 200 μg/mL + 0.12 μM. Data are presented as mean ± SD of three independent experiments (*n* = 3). #### *p* < 0.0001, ## *p* < 0.01 as compared to control, $$ *p* < 0.01, $$$$ *p* < 0.0001 as compared to MGO+H, &&&& *p* < 0.0001 as compared to MGO and **** *p* ˂ 0.0001 as compared to MGO+C.

**Figure 4 pharmaceuticals-15-00347-f004:**
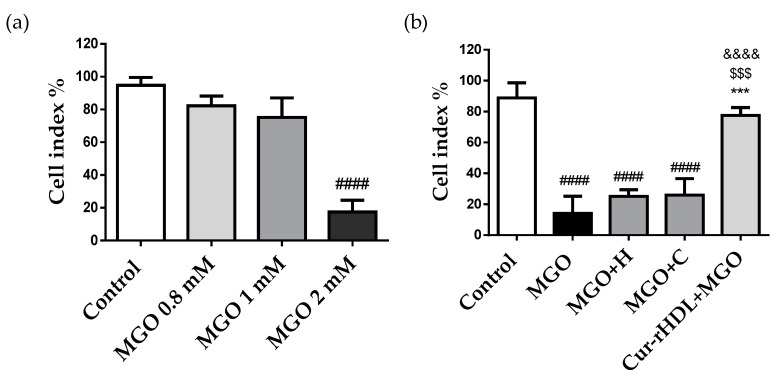
Effect of curcumin-enriched rHDLs on the cerebral endothelial cell monolayer integrity assessed by the measurement of electrical impedance. (**a**) The cells were treated with different MGO concentrations (0.8–2 mM). (**b**) The cells were pre-treated with HDL (80 μg/mL), curcumin (0.048 μM), curcumin-enriched rHDLs (Cur-rHDLs-80 μg/mL + 0.048 μM) for 1 h before the addition of 2 mM MGO. The cell index was measured as electrical impedance generated by cell attachment and proliferation detected by electrodes present at the bottom of the plate. Data are presented as mean ± SD of three independent experiments (*n* = 3). #### *p* ˂ 0.0001 as compared to control, $$$ *p* ˂ 0.001 as compared to MGO+H, *** *p* ˂ 0.001 as compared to MGO+C and &&&& *p* ˂ 0.0001 as compared to MGO.

**Figure 5 pharmaceuticals-15-00347-f005:**
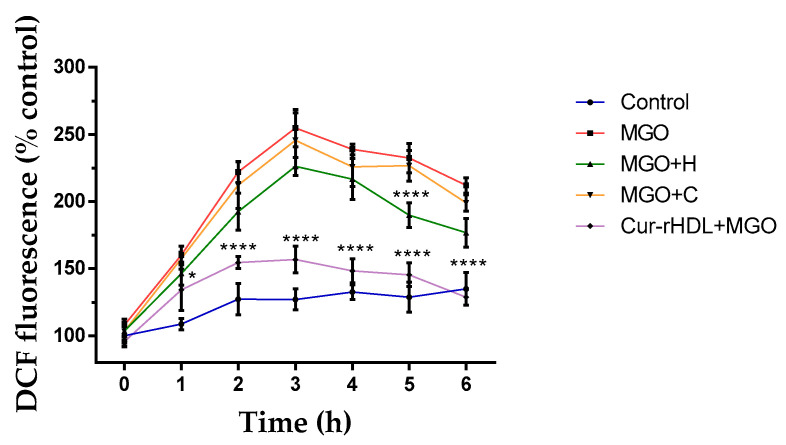
Effect of curcumin-enriched rHDLs on intracellular ROS production induced by MGO in cerebral endothelial cells evaluated by the DCFH-DA assay. The cells were pretreated with rHDLs (H-80 μg/mL), curcumin (C-0.048 μM), and curcumin-enriched rHDLs (H+C-80 μg/mL + 0.048 μM) for 1 h before the addition of 2 mM MGO for 1–6 h. Intracellular ROS were quantified by the measurement of DCFH-DA fluorescence. Data are rpresented as mean ± SD of three independent experiments (*n* = 3). * *p* ˂ 0.05, **** *p* ˂ 0.0001 as compared to MGO.

**Figure 6 pharmaceuticals-15-00347-f006:**
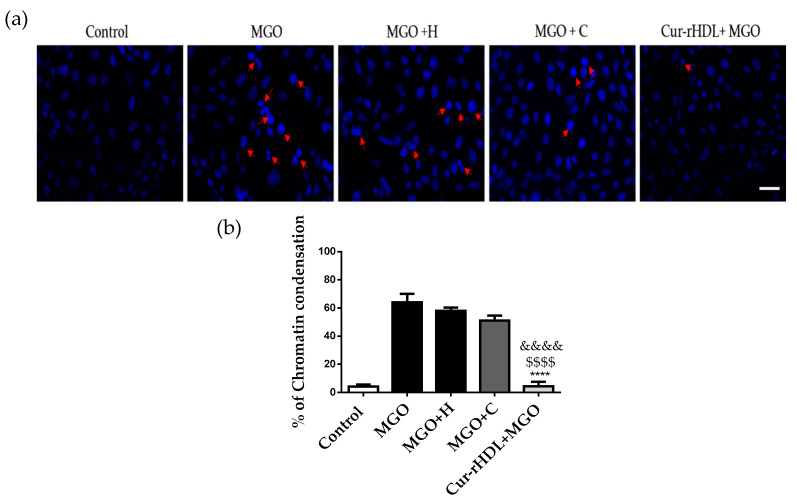
Effect of curcumin-enriched rHDLs on chromatin condensation induced by MGO in cerebral endothelial cells by DAPI staining. (**a**) DAPI staining images; the cells were treated with rHDLs (H-80 μg/mL), curcumin (C-0.048 μM), and curcumin-enriched rHDLs (Cur-rHDL-80 μg/mL + 0.048 μM) before the addition of 2 mM MGO for 24 h. Red arrows indicate a typical example of chromatin condensation. (**b**) The results were obtained by counting the number of cells with condensed chromatin. Scale bar: 60 μm. Data are presented as mean ± SD of three independent experiments (*n* = 3). $$$$ *p* ˂ 0.0001 as compared to MGO+H, **** *p* ˂ 0.0001 as compared to MGO+C and &&&& *p* ˂ 0.0001 as compared to MGO.

**Figure 7 pharmaceuticals-15-00347-f007:**
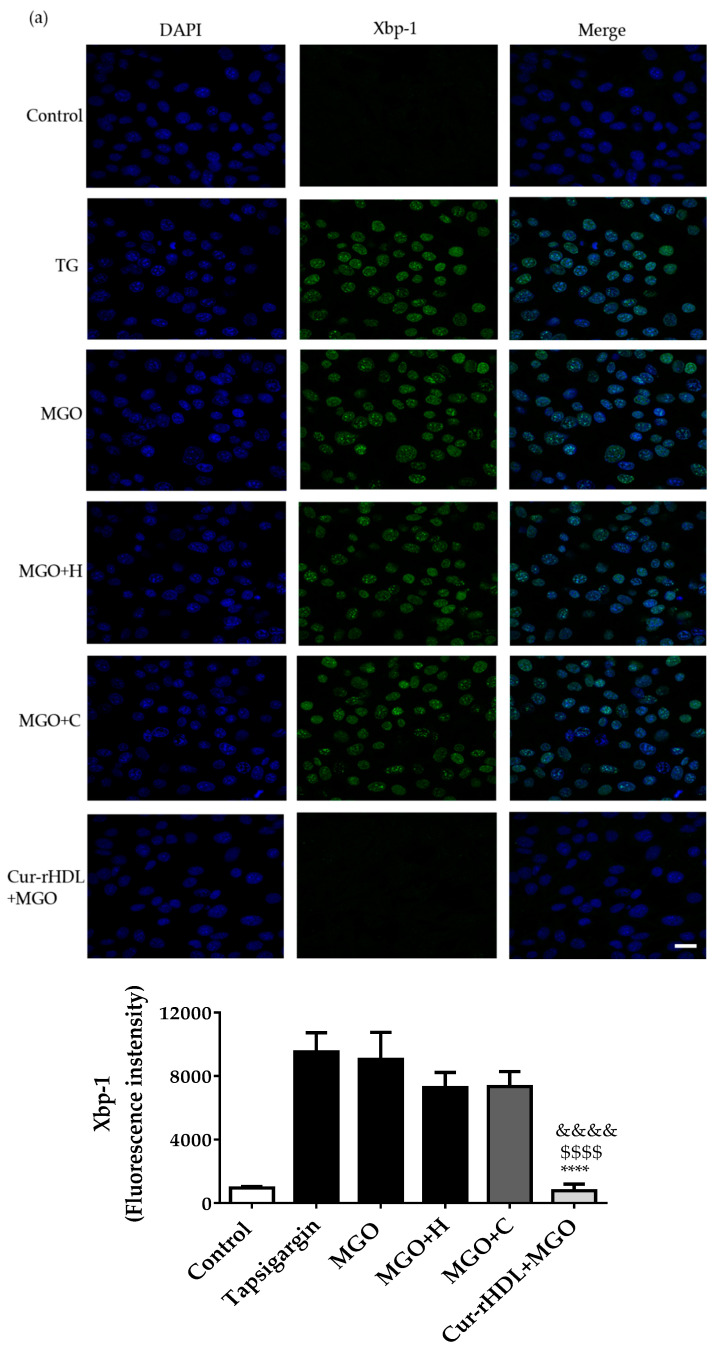
Effect of curcumin-enriched rHDLs on ER stress induced by MGO in cerebral endothelial cells assessed by immunohistofluorescence analysis. (**a**) Confocal microscopy images of the nuclear translocation of Xbp-1, (**b**) ATF-4, and (**c**) GRP 78 markers. The cells were treated with 1 μg/mL of thapsigargin, used as positive control for ER stress (TG), 2 mM methylglyoxal (MGO), pretreated with rHDLs (H, 80 μg/mL), curcumin (C, 0.048 μM), or curcumin-enriched rHDLs (Cur-rHDL, 80 μg/mL + 0.048 μM) before the addition of 2 mM MGO for 6 h. Scale bar: 24 μm. Data are presented as mean ± SD of three independent experiments (*n* = 3). $$ *p* ˂ 0.01, $$$$ *p* < 0.0001 as compared to MGO+H, ** *p* < 0.01, **** *p* < 0.0001 as compared to MGO+C and && *p* < 0.01, &&&& *p* < 0.0001 as compared to MGO.

**Figure 8 pharmaceuticals-15-00347-f008:**
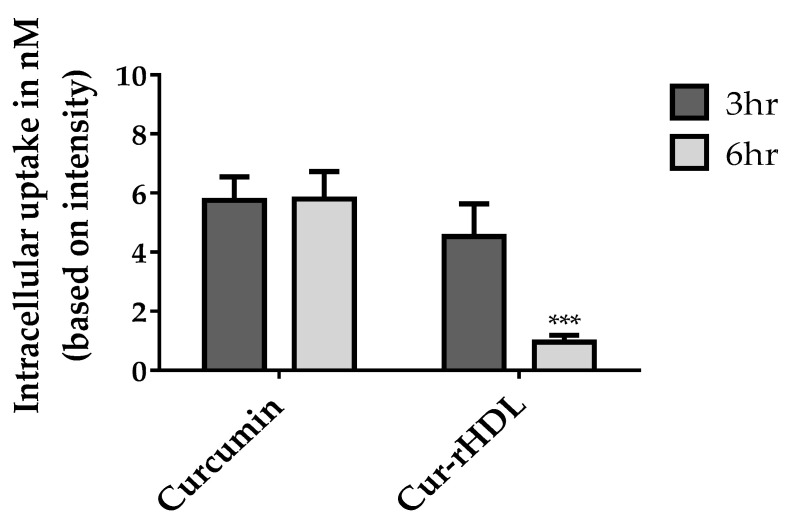
LC–MS/MS analysis of curcumin uptake in cerebral endothelial cells after incubation (3–6 h) with curcumin-enriched rHDLs or curcumin alone. The samples were prepared and analyzed by LC–MS/MS as described in the materials and methods. Data are presented as ±SD of three independent experiments (*n* = 3). *** *p* ˂ 0.001 as compared to curcumin.

**Figure 9 pharmaceuticals-15-00347-f009:**
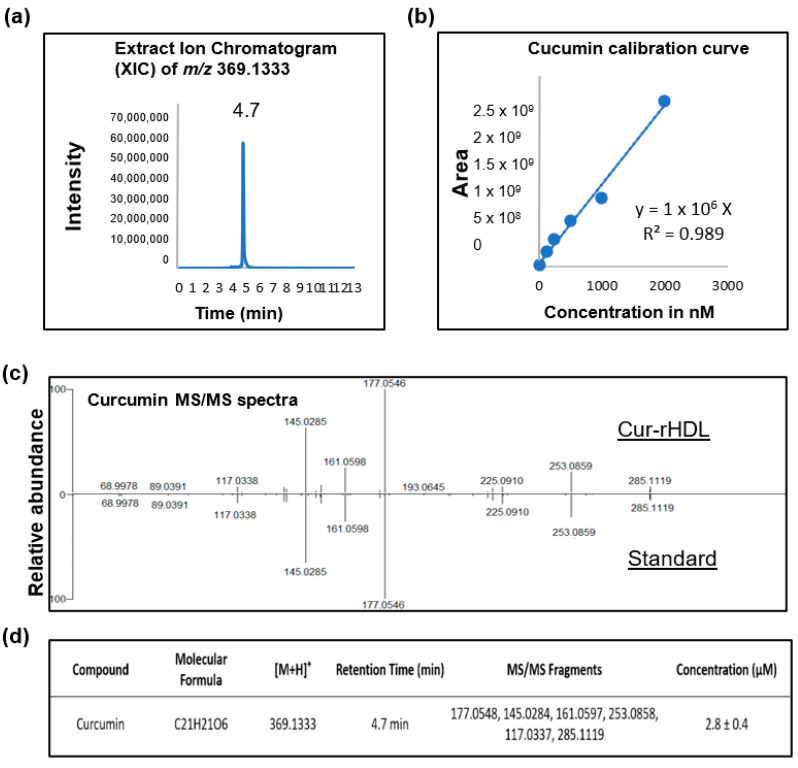
Quantification of curcumin in curcumin-enriched rHDLs by LC–MS/MS analysis. (**a**) Representative ion chromatogram of curcumin, (**b**) Calibration curve used to quantify curcumin, (**c**) LC–MS/MS analysis spectrum of curcumin in the Cur-rHDL sample vs. the curcumin standard and (**d**) Combined LC–MS/MS analysis details of curcumin quantification.

## Data Availability

Data is contained within the article.

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
