# Peer review of "ApoA-I Nanoparticles as Curcumin Carriers for Cerebral Endothelial Cells: Improved Cytoprotective Effects against Methylglyoxal"

_pharmaceuticals, 2022, doi:10.3390/ph15030347_

Round 1

Reviewer 1 Report

Apolipoprotein A1 --> apolipoprotein A-I

ApoA1 --> apoA-I

Ref: means Cat No ?

Please spell out abbreviations for readers as below

FWHM, ESI, DMSO, ACN

The  unit is confused. It should be used as the same way.

Please fix one between ml and mL, μl and μL

Pleaseremove space in LC- MS/MS

Please revise FLUOstar Optima, Bmg Labtech, and Cambridge, UK

San Diego, CA, United States should be changed into USA

Many spaces sholud be removed,  for example, Cell signaling technology- 11815S

Massachusetts  --> MA

Please explain 500 μL of MeOH- HCl (200 mM)/well

p-value  --> p value

Subtitle is too long

3.1 MGO cytotoxicity and effect of HDLs and curcumin on cerebrovascular endothelial 222
cell cytotoxicity induced by MGO

0.8 mM-2 mM  --> 0.8-2 mM

In many parts, punctuations, spaces, and italicized  fonts such as p value should be fixed in entire text

Reviewer 2 Report

The manuscript reported the synergistic effect of cur-rHDL nanoparticles on bEnd.3 cytotoxicity, reactive oxygen specie (ROS) production, chromatin condensation, endoplasmic reticulum (ER) stress, and endothelial barrier integrity by impedancemetry. I think this study is well designed and the results are informative. Authors should consider the following concerns to improve the quality of this study.

  1. If possible, please provide some characterization information of Cur-rHDL nanoparticles, such as TEM images and particle size distribution. Article “Fe-Curcumin Nanozyme-mediated ROS Scavenging and Anti-inflammation for Acute Lung Injury” may be helpful for authors.
  2. In figure 1 (b), the spectrum of curcumin in Cur-rHDL sample and curcumin standard is same. Please make sure if there were some mistakes. Moreover, suggest marking which part is the spectrum of curcumin in Cur-rHDL and which one of curcumin standard.
  3. In figure 1 (c), the R2 of calibration curve is less than 0.99. Suggest verifying this calibration curve repeatedly.
  4. For the confocal microscopy images of ATF-4 (figure 7 (b)), the DAPI fluorescence intensity of Cur-rHDL + MGO group is obviously weaker than other groups. Generally, the parameter setting of confocal microscopy for each group should be consistent.
  5. Line 435-437, please provide related experimental data or citation to demonstrate “decrease of transendothelial electrical resistance by MGO was associated with altered tight junction proteins”.
  6. Line 466-469, “Previous studies showed that curcumin encapsulated……”, please provide the relate citation.
